# A Brief Overview of Electrochromic Materials and Related Devices: A Nanostructured Materials Perspective

**DOI:** 10.3390/nano11092376

**Published:** 2021-09-13

**Authors:** Aleksei Viktorovich Shchegolkov, Sung-Hwan Jang, Alexandr Viktorovich Shchegolkov, Yuri Viktorovich Rodionov, Anna Olegovna Sukhova, Mikhail Semenovich Lipkin

**Affiliations:** 1Department of Chemical Technologies, Platov South-Russian State Polytechnic University (NPI), 346428 Novocherkassk, Russia; lipkin@yandex.ru; 2Department of Civil and Environmental Engineering, Hanyang University ERICA, Ansan 15588, Korea; 3Department of Technology and Methods of Nanoproducts Manufacturing, Tambov State Technical University, 392000 Tambov, Russia; Energynano@yandex.ru; 4Department of Mechanics and Engineering Graphics, Tambov State Technical University, 392000 Tambov, Russia; rodionow.u.w@rambler.ru; 5Department of Nature Management and Environment Protection, Tambov State Technical University, 392000 Tambov, Russia; apil1@yandex.ru

**Keywords:** electrochromic materials, nanostructured electrochromic materials, electrochromism, color, “Smart Windows”, transition metal oxides (TMO), nanomaterials, graphene oxide (GO), reduced graphene oxide (rGO)

## Abstract

Exactly 50 years ago, the first article on electrochromism was published. Today electrochromic materials are highly popular in various devices. Interest in nanostructured electrochromic and nanocomposite organic/inorganic nanostructured electrochromic materials has increased in the last decade. These materials can enhance the electrochemical and electrochromic properties of devices related to them. This article describes electrochromic materials, proposes their classification and systematization for organic inorganic and nanostructured electrochromic materials, identifies their advantages and shortcomings, analyzes current tendencies in the development of nanomaterials used in electrochromic coatings (films) and their practical use in various optical devices for protection from light radiation, in particular, their use as light filters and light modulators for optoelectronic devices, as well as methods for their preparation. The modern technologies of “Smart Windows”, which are based on chromogenic materials and liquid crystals, are analyzed, and their advantages and disadvantages are also given. Various types of chromogenic materials are presented, examples of which include photochromic, thermochromic and gasochromic materials, as well as the main physical effects affecting changes in their optical properties. Additionally, this study describes electrochromic technologies based on WO_3_ films prepared by different methods, such as electrochemical deposition, magnetron sputtering, spray pyrolysis, sol–gel, etc. An example of an electrochromic “Smart Window” based on WO_3_ is shown in the article. A modern analysis of electrochromic devices based on nanostructured materials used in various applications is presented. The paper discusses the causes of internal and external size effects in the process of modifying WO_3_ electrochromic films using nanomaterials, in particular, GO/rGO nanomaterials.

## 1. Introduction

Modern technology has a number of negative effects, such as atmospheric pollution, global warming, the reduction of fossil resources, etc. Therefore, one of the most important tasks in the world is to improve energy efficiency and energy savings. To this end, it is necessary to create new materials in a variety of sectors, including engineering, agro-industry, building construction, electronics manufacturing, etc., primarily with the aim of using new technologies and “smart” materials.

Functional materials are dependent on their initial state and properties, as well as on the energy and external effects applied to the material. “Smart” materials have more than one functional state, depending on the impacting impulse, which can change over time [1].

Electrochromic materials (EC) are materials that are able to change color under the influence of an electric field. EC are of great interest, both from the scientific point of view and with respect to their application in various technical systems, including as the basis for the creation of electrochromic devices (ECD) with low power requirements, such as [2,3,4]:-“Smart Windows”;-Displays;-Reflective blinds;-Variable reflection mirrors;-Sensors.

The main purpose of ECD is protection against light in the visible wavelength range (380–780 nm). ECD include an electrochromic coating in the form of the EC film and a counter electrode placed in an electrolyte (ionic conductor), which is located between transparent conductive electrodes—ITO (In_2_O_3_-SnO_2_) or FTO (SnO_2_-SnF). The principle of ECD operation is the transformation of optical light flux and the modulation of the coefficient of light reflection/transmission, resulting in an electrochemical reaction, i.e., the “Smart Window” effect.

Thus, “Smart Window” technology allows savings due to use of smaller amounts of energy for air conditioning in summer, as well as for heating in winter; an average of more than 30% compared to conventional windows.

The purpose of this review is to systematize and summarize the data on organic, inorganic and nanostructured electrochromic materials and related devices over the past 50 years.

## 2. “Smart Windows”

There are chromogenic materials [3], better known as “smart” materials, that are currently experiencing great popularity. These materials modulate reflected or diffused light by means of physical effects of different types. The “Smart Window” based on chromogenic materials is widely used in architecture, cars (rear-view mirrors and intelligent window tinting), and aircraft illuminators (Boeing 787 Dreamliner) [3,4,5], while translucent structure technology [6] must also be mentioned. Chromogenic materials change color and transparency. The following types of chromogenic materials can be distinguished: electrochromic materials (EC) (external conditions—electric field); photochromic materials (PhC) (external conditions—light); thermochromic materials (ThC) (external conditions—heat); gasochromic materials (GhC) (external conditions—gas); polymer-dispersed liquid crystals (PDLC) and liquid crystal dispersions (LCD); SPD—suspended particles device are placed between the two electroconductive coatings. These materials can serve as a basis for “Smart Window” technologies [7,8,9,10,11], which are shown in Figure 1.

The structures of SPD and PDLC windows [6,10,11,12] are shown in Figure 2. SPD technology (Figure 2a) uses suspended particles to modulate light transmission, arranging themselves in an alternating current field, and the film becomes transparent. In the absence of the electric field, the SPD window acquires color and absorbs light. The SPD window is similar in structure to the PDLC window (Figure 2b), apart from the fact that in the absence of an electric field, the film becomes semi-transparent.

Electrochromic windows (ECW) control the transmission of light in the visible spectrum and switches between tinted and transparent/semi-transparent states in response to low voltage signal (Figure 3). ECW create a comfortable indoor environment; moreover, they have lower power consumption in comparison with other chromogenic devices [13]: the ECW modulates reflected light under the control voltage, and in the absence of the control voltage, modulation of the transmitted light occurs [2,3,4,7].

The advantages of electrochromic technologies are as follows [14]:-electric energy is required only during mode switching;-low activation voltage (1–5 V);-a wide variety of “Smart Window” tints (blue, grey, brown, etc.);-in the bleached state, electrochromic devices have a transparency level of 50–70%, in the colored state—10–25%.

Table 1 shows the basic advantages and shortcomings of chromogenic materials used in “Smart Windows”.

Electronic devices emit high levels of electromagnetic radiation (EMR) in a wide frequency range, leading to electromagnetic pollution, which negatively influences biological objects and causing electronic device dysfunction [15]. Considering the fact that electromagnetic radiation basically penetrates through glass surfaces, the problem of creating a universal electrochromic film capable of absorbing or reflecting electromagnetic radiation is becoming relevant.

## 3. Electrochromism and Electrochromic Materials: Classification and Applications

Chromism (from ancient Greek χρῶμα (“color”)) is a phenomenon of material color change under the influence of physical factors, such as electric field, heat, light or pressure [16].

At the end of the 1960s, scientist S. K. Deb discovered the phenomenon called electrochromism [17]. This newly discovered phenomenon belongs to the sphere of electrochemistry and physics [18]. S. K. Deb described a new electrophotographic system consisting of WO_3_ thin film and a thin-film photoconductive layer placed between two electrodes. When this composite structure was subjected to an electric field, an optical projection appeared. After subsequent modulation in the photoconductive layer, the oxide layer acquires the same color, and a visible image appears [17,18,19].

Since the middle of the 1970s, electrochromism has been considered to be a physical phenomenon associated with a reversible change in transparency or color under the influence of an electric field or electric current [19,20]. Electrochromism is traditionally defined as a reversible change in optical properties (transparency and/or reflectivity) during the oxidation–reduction reaction [21,22,23]. In some cases, there are more than two degrees of oxidation, and the material is capable of showing several colors depending on the current degree of oxidation (polyelectrochromic materials) [21]. Modern science uses a broad definition of electrochromism, including materials and devices used for the optical modulation of radiation in the visible and microwave ranges. Ref. [24] focuses on the problem of developing electrochromic displays that should replace LED and liquid crystal displays. In 1985, Svensson and Granqvist proposed using electrochromic materials in “Smart Windows” [14], and thus the term “Smart Window” appeared. The electrochromic reaction can be described by the electrochemical equation in oxidized form:O + xe^−^ + Cation ↔ Reduced form, R(1)

Applications of electrochromism include:-Control of energy transfer in different environments, for example, filtering solar radiation using “Smart Window” devices [25,26,27]. Fast mode switching (colored/bleached) is not required, but the device should be capable of filtering both visible and near-infrared radiation. Moreover, the transparency of the window packages must be at least 70%.-Color displays [22,24], for example, advertising boards. The requirements are as follows: fast mode switching, color scheme varies only in the visible area. Moreover, color contrast should be high enough, transparent mode is not required.-Mirror light modulators [7], for example, antiglare mirrors for cars. Fast mode switching and high transparency are not required.

### 3.1. Classification of Electrochromic Materials

There are several inorganic and organic EC that change their optical properties (transparency, color) during oxidation–reduction [28,29,30,31,32]. Switching between oxidation and reduction states leads to color formation, i.e., formation of new spectral peaks in the visible area. Inorganic EC include transition metal oxides (TMO) from groups IV-VI [32], and hexacyanometallates (Prussian blue). Organic EC include viologens, conjugated conductive polymers (polypyrrole, polythiophene, polyaniline and their derivatives, metal polymers, metal phthalocyanines) [33,34]. The viologen family (4,4′-dipyridinium compounds) has a general chemical formula as shown in Figure 4, where R may be an alkyl, cyclo-alkyl or other substitute, and X corresponds to halogen 4,4′-dipyridium compounds, because they turn a deep blue-purple on reduction [30]. The viologen ion as shown in Figure 4a can have a two-step reduction, i.e., a one-electron or a two-electron reduction. The general structure for viologens modifying the titania surface is shown in Figure 4c. Table 2 presents a list of the most popular EC.

Table 2 shows a general classification of EC.

In general, organic EC, possessing color-changing abilities, exhibit faster response times and higher staining efficiencies than inorganic ones, but have a low UV protection index and show lower electrochemical stability. Therefore, mainly organic EC materials are used in electronic non-emissive displays [24,28]. Inorganic EC materials show high chemical stability and cyclicity, which makes them suitable for “Smart Windows” and large-scale data displays [35].

Electrochromic materials are classified according to their solubility and according to their redox states [29,30]. Classification of EC was introduced by I. F. Chang in 1975 [36]. According to this classification, there are three types of EC solubility in redox states [29]:(1)Type I EC materials, such as viologen, heptyl, etc., are soluble in both their reduced and oxidized states.(2)Type II EC materials are soluble in their colorless redox state but form a solid film on the electrode surface.(3)Type III EC materials are solids in both redox states, and they form an insoluble film on the electrode surface. Type III materials include groups IV, V transition metal oxides (TMO), conductive polymers, Prussian blue and metal polymers. Three types of mechanism for changing color/transparency (according to I. F. Chang) are presented in Figure 5.

The electrochromic reaction can be described by the following equation:EC^n^ + yCE^m^_(bleached)_ ↔ EC^n−a^_(coloured)_ + yCE^(m+a)/y^(2)

Table 3 contains examples of each type of EC.

Type I and Type II EC are self-erasing, since an electrical current is required to maintain the colored state, i.e., after the power is turned off, the ECW loses its color. Type III ECW (battery-powered) remain colored for some time after the voltage is removed. Electrochromic technologies make it possible to modulate the optical properties, such as color, light transmission coefficient *T(λ)*, reflection coefficient *R(λ)*, and absorption coefficient *A(λ)*, of materials according to Kirchhoff’s law [44]:*R*(*λ*) + *A*(*λ*) + *T*(*λ*) = 1(3)

All of these optical processes (Figure 6) are characterized by the EC transmittance *T(λ)*, absorption *A(λ)* and reflectance *R(λ)*, which indicates the proportion of the incident light intensity that passes through, is absorbed by, or is reflected by the EC.

Electrochromic properties depend on the electrochromic film structure; thus, different EC have different absorption spectra, and, consequently, differ in color.

### 3.2. Organic EC

Organic films, such as conductive polymers, have multiple colored states, possess high optical contrast, and exhibit fast response time and high staining efficiency [45,46,47,48,49].

Electrochromic behavior is observed in conjugated pyridine derivatives such as viologens (Figure 7), which exhibit high cyclicity, low operating potential and other valuable properties [50,51]. Viologens exist in solid crystalline form and in powder form. The name ‘viologen’ alludes to violet, one color it can exhibit (Figure 7).

Viologens are used in RGB (red, green, blue) devices (Figure 8), which reproduce three main colors, red, green and blue, although research in this area is not yet well developed. Modern technologies require the use of multicolor EC, which, in turn, necessitates the creation of new functional composites [50].

The advantages of organic EC include compatibility with flexible substrates, low production cost and the possibility of adjusting their synthetic material properties.

Phenylenediamine (PD) derivatives are of interest due to their stable electrochemical reactions at the anode [52]. It is interesting to note that neutral arylamine is often colorless (it mostly absorbs UV light), but in redox states, it exhibits vivid color (Figure 9). Phenylenediamines exhibit modulated visible absorption properties and high redox stability, which makes them suitable for RGB devices [53].

The color-changing abilities of conductive PEDOT polymers [54] are useful in electrochromic non-emissive displays (Figure 10).

Heterocyclic aromatic compounds (Figure 11), such as thiophene, aniline, furan, carbazole, azulene and indole [55,56], can be oxidized chemically or electrochemically to form anion-doped polypyrrole (PPy), polythiophene (PT) or polyaniline (PANI), poly(3-methylaniline) (MEPA), poly(3-methyl-thiophene) (P3MTh), and poly(3-methylpyrrole) (P3MPy). A change in the redox state (oxidized conductive state, reduced non-conductive state, neutral state) leads to changes in color caused by significant changes in the visible and near-infrared absorption spectra that vary depending on the degree of oxidation/reduction Switching between polymer films in their colored (reduced) and uncolored (oxidized) states changes their color from yellow to orange, red, purpuric, dark blue, green, light blue, and black [57].

Table 4 shows conductive polymers obtained by the oxidation of monomeric aromatic compounds (neutral and oxidized states).

The shortcomings of polymer films include their low electrochemical stability and, consequently, their low oxidation number [28,29,30]. The addition of inorganic materials improves the properties of electrochromic conductive polymers [58]. Inorganic materials improve the staining efficiency and reduce the switching time, but do not affect the electrochemical properties of the polymer; therefore, the problem of improving polymer electrochemical stability is still relevant. Electrochromic films, such as WO_3_, Nb_2_O_5_, NiO, are preferable due to their high stability and durability.

### 3.3. Transition Metal Oxides

Inorganic materials include a large group of EC, mostly the TMO Me_x_O_y_ (Figure 12). The most common TMO [32,59], such as molybdenum (VI) oxide, vanadium (V) oxide, niobium (V) oxide, iridium (III) oxide, tungsten (VI) oxide, are in the form of an octahedron MeO_6_ (Figure 13). The crystal structure of CWO_3_ perovskite shown in Figure 14.

In the mentioned structures, electrochromic effects occur due to electron–ion separation. As a result, metal atoms are introduced into TMO, and the valence electrons move to the d-levels of the transition metal ion, reducing it. Evidently, the injected ions should possess a high diffusion coefficient and a high solubility in the lattice of TMO [18,20].

There are several highly efficient TMO (IrO_2_ [60], MoO_3_ [61], NiO [62], TiO_2_ [63], WO_3_ [42,59]) that are colorless in the oxidized state and colored in the reduced state (cathodic EC, color change is induced by ion injection). Inorganic compounds that are colorless in their reduced state and colored in their oxidized state are called anodic EC (color change is induced by ion extraction).

Vanadium oxides [64] exhibit hybrid features, and ECD usually contain two EC films [32,59]; therefore, it would be relevant to simultaneously use a cathodic oxide (for example, Mo or Nb) and an anodic oxide (for example, Ni or Ir) [61,63].

EC exhibit polychromism [65], for example, amorphous Nb_2_O_5_ is brown in its colored state, while crystalline Nb_2_O_5_ acquires a blue color; WO_3_ is blue in its colored state, while TiO_2_ obtains its color (blue or grey) as a result of ion injection (H^+^ or Li^+^, respectively). The most investigated cathodic EC is WO_3_ [66]. The color change mechanism has still not been sufficiently investigated, but most scientists agree that the extraction and injection of electrons and metal cations (Li^+^, H^+^, Na^+^, K^+^, etc.) play a crucial role in color change. NiO and IrO_2_ are the most popular anodic EC. High concentrations of cations in the electrolyte, which is an ion conductor, significantly affect the electrochromic properties of the TMO, such as switching time, cyclicity and staining efficiency.

The majority of TMO have a band gap of 1–5 eV (Figure 15), and therefore occupy an intermediate position between semiconductors and dielectrics [67]. EC behavior is dependent on TMO structure. It should be noted that structural and impurity defects directly affect the properties—particularly the physicochemical properties—of the EC under study.

The optical band gap can be calculated according to Equation (4) [37]:(4)$$\alpha \mathrm{hv}=A{\left(\mathrm{hv}-E_{g}\right)}^{n}$$
where α is the absorption coefficient, which can be measured by the ultraviolet spectrophotometer; h is the Planck constant; v is the light frequency; A is a proportionality constant; E_g_ is the optical band gap; n is a number that is ½ for the direct band gap semiconductor and 2 for the indirect band gap semiconductor.

The E_g_ of the WO_3_ films decreased from 3.62 eV to 3.30 eV when the annealing temperature was increased. In addition, the E_g_ of the colored WO_3_ films was less than that of the bleached WO_3_ films [38]. The different band gap demonstrates that the conductivity of the WO_3_ film is enhanced with decreasing E_g_, while the high conductivity increased the electrochromic response time.

The transparency of inorganic EC with high staining efficiency varies in response to the low-voltage signal. WO_3_ and NiO (Table 5) have a staining efficiency of ~40 sm^2^∙C^−1^, while for organic EC films, such as PEDOT, this value is more than 100 sm^2^∙C^−1^ [32,39]. Actually, TMO have a high physical and chemical stability.

TMO belong to type III materials, according to I. F. Chang’s classification. Both anodic (A) and cathodic (C) reactions are possible, depending on the redox state of the electrochromic film. Table 6 describes the electrochemical anodic and cathodic reactions of certain oxides.

### 3.4. WO_3_ Electrochromic Films

Tungsten (VI) oxide (WO_3_) is the most universal EC, and its electrochromic properties were first described by S. K. Deb in 1969 [17]. This oxide is still widely investigated [32,40]. High functionality, high staining efficiency, high contrast, high chemical stability, and long life cycle are all features that make tungsten (VI) oxide useful in practice [41,43]. WO_3_ electrochromic films exhibit a deep blue color, preserve their color for some hours after the voltage is removed (electrochromic memory), and demonstrate high cyclic stability in comparison to other TMO [32]. The electrochromic mechanism of WO_3_ film is shown in Figure 16.

WO_3_ films have different colors depending on x. At low values of x, the film is colored blue, and at high values of x, it has either a red or golden tint. These phenomena are associated with the fact that, firstly, WO_3_ is partially reduced to the oxidation state V+, and secondly, the addition of the Li ^+^ cation occurs; all this leads to changes in the band gap and, as a consequence, in the light transmittance of the TMO.

At the same time, the molecular reaction in WO_3_ films can be described as follows [68]:
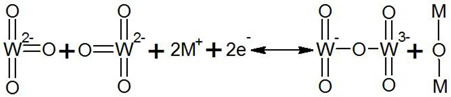
(5)

In [69], it was shown that the electrochemical reaction at the WO_3_/electrolyte interface plays an imperative role in the electrochromic performance of WO_3_ electrodes, and the lithium-ion transformation mechanism at the WO_3_/electrolyte interface was demonstrated, wherein the states are replaced from one phase to another.

The high efficiency of amorphous WO_3_ films [40,70] manifests in a reversible switch from transparent to dark blue during electrochemical redox reactions (Figure 16). Electrochromic properties, such as staining efficiency, and switching time are dependent on the atomic structure, nanoparticle size, pore size and absorption properties [71,72]. O. F. Schirmer suggested that the optical absorption phenomenon in WO_3_ films was due to small polaron (SP, charged and polarized quasiparticles) transitions from W^(V)^ ions to W^(VI)^ ones. In [42,69,73], the light absorption mechanism in amorphous WO_3_ was is described as the interval optically induced transfer of 5d1-electron of the W^(V)^ ion (A) to the adjacent empty 5d0-orbital of the W^(VI)^ ion (B):(6)$$W^{\left(V\right)}\left(A\right)+W^{\left(VI\right)}\left(B\right)\overset{h\nu }{\to }W^{\left(VI\right)}\left(A\right)+W^{\left(V\right)}\left(B\right)$$
where A and B represent tungsten sublattice knots.

This phenomenon was studied using X-ray photoelectron spectroscopy (XPS) and electron spin resonance (ESR) spectroscopy [74,75]. The WO_3_ films showed high absorption in the near-infrared region due to polaron absorption [76]. Activated WO_3_ films are characterized by a wide absorption band with a maximum of 0.9–1.46 eV, depending on the film properties [73]. Figure 17 shows the optical transmission spectra of WO_3_ (Figure 17a) and WO_3_/GO (Figure 17b) films upon coloring and bleaching.

The optical properties of WO_3_ thin films depend on their structure (crystalline, polycrystalline, amorphous or hybrid). Colored and colorless states of WO_3_ films are not symmetric. Switching from transparent to colored states, polycrystalline WO_3_ films exhibit reflective properties, and amorphous WO_3_ films exhibit absorption properties. The switching time depends on WO_3_ film density and on electrolyte concentration. Low-density films with high-concentration electrolytes demonstrate the fastest switching speed [78].

Nowadays, the importance of WO_3_ films has grown [79,80] due to their use in “Smart Windows”, which smartly regulate indoor solar radiation by changing their optical transmittance, contributing to a significant reduction in a building’s energy consumption (as a result of the optimization of air conditioning consumption) and helping to create comfortable indoor environments [81]. However, despite all the advantages of WO_3_ films, their life cycle is not very long: continuous switching between colored and colorless states causes irreversible structural changes that affect their optical and electrical properties, ultimately leading to material degradation, the so-called “aging” effect [82]. Therefore, the task of increasing the life cycle of WO_3_ films involves the development of new nanomaterials and/or the improvement of existing materials through the use of modificatory additives, as well as the obtained improvement of WO_3_ film technologies [83,84,85,86].

## 4. ECD (Electrochromic Device) Structure

EC are able to reversibly change their optical properties through the application of an electrical voltage, making them suitable for ECD, such as displays [30], electrochromic “Smart Windows” [16], anti-glare rear mirrors [19], and sensors [87].

ECD structures usually include transparent conductors, electrochromic layers, and ion conductors (Figure 18).

ECD for architectural applications include thin EC films placed between two glass panels, as shown in Figure 19 and Figure 20.

Cycle stability is an extremely important aspect in the performance of electrochromic devices. In a recent study [88], ECD were reported to have obtained a superior long-term cycling stability of over 10,000 cycles. This manuscript is recommended for its review of some reports of devices with high long-term stability. In [89], a strategy was presented involving an all-in-one self-healing electrochromic material, TAFPy-MA, which was used for the fabrication of a high-reliability, large-scale and easy-to-assemble smart electrochromic window. The all-in-one self-healing electrochromic material was able to carry out in situ redox reactions with the Li^+^ ions. The Diels-Alder cross-linking network structure was able to heal the cracks, improving the reliability of the electrochromic layer. Great ion diffusivity (1.13 × 10–5 cm^2^ s ^−1^), rapid color switching (3.9/3.7 s), high coloration efficiency (413 cm^2^ C^−1^), excellent stability (sustain 88.7% after 1000 cycles) and reliability (crack can be healed in 110 s), large-scale “Smart Windows” (30 × 35 cm^2^) were achieved using this all-in-one electrochromic material, and these exhibited fascinating and promising features for a wide range of applications in buildings, airplanes, etc.

Electrochromic films change their color as a result of electrochemical oxidation/reduction reaction associated with ion transfer, which involves the use of an additional coating for the storage and transport of ions. Many companies offer “Smart Window” solutions; energy-saving “Smart Window” technology is available on the market [4].

Depending on the purpose, ECD may contain materials with different characteristics and properties. Figure 21 presents the classifications of ECD.

### 4.1. Substrate

Transparent ECW substrates include glassy (Figure 22) or transparent polymers (Figure 22), such as polyethylene terephthalate (PET), polyvinyl butyral (PVB) and polyethylene naphthalate (PEN). Glass substrates are more common due to their greater transparency and their chemical stability, which makes them suitable for the production of “Smart Windows”. In turn, polymer substrates allow the production costs of ECD to be reduced [90,91,92,93].

### 4.2. Transparent Conductive Electrode

The electrical resistivity and the light transmission coefficient are the most important properties of transparent conductive electrodes (layers). An electrode should possess high electrical conductivity in order to form the electric field required for ECD. Transparent conductive electrodes include metal-based and oxide-based electrodes, but the electrode properties should not affect the transmission properties of the electrochromic windows. Indium-tin oxide (ITO) electrodes (indium (III) oxide and tin (IV) oxide) are among the best transparent electrodes to have been investigated ((In_2_O_3_)0.9-(SnO_2_)0.1: 90% and 10%) [90], possessing high electrical conductivity (~104 S∙sm^−1^) and low optical absorption (band gap ~4 eV, refractive index −1.9), making it preferable to fluorine-doped tin oxide (FTO). Transparent ITO electrode contains different numbers of doped Sn atoms, and consequently, free electron density varies [94].

### 4.3. Electrochromic Layer

EC films reversibly change their optical properties, switching between transparent, semi-transparent and colored states, modeling solar radiation and thus ensuring reliable ECW operation. EC films (layers) can be divided into three different types according to their color schemes [32]:-EC film exhibiting one color, for example, transition metal oxides, Prussian blue [31];-EC film exhibiting two colors, for example, polythiophene [28];-EC film exhibiting multiple colors, for example, poly (3,4-propylenedioxypyrrole) [29].

### 4.4. Electrolyte (Ion Conductor)

Electrolytes can be classified into liquid, gel and solid electrolytes [32]. Liquid electrolytes are dissolved ions. Such electrolytes provide high ionic mobility. Polymer electrolytes are the most suitable for EC devices, as they provide a long circuit break and uniformity of coloration [95].

Electrochromic device electrolytes are ionic materials that possess ionic conductivity. Electrochromic device electrolytes should satisfy the following requirements [77]:-compatibility with anodic and cathodic materials;-high ionic conductivity;-no electron transfer between electrochromic layers;-high transparency without scattering effect.

In [96], a novel Zn–Prussian blue (PB) system was reported for aqueous electrochromic batteries. By using different dual-ion electrolytes with various cations (e.g., Zn^2+^–K^+^ and Zn^2+^–Al^3+^), the Zn–PB electrochromic batteries demonstrated excellent performance. We showed that the K^+^–Zn^2+^ dual-ion electrolyte in the Zn–PB configuration endowed a rapid self-bleaching time (2.8 s), high optical contrast (83% at 632.8 nm), and fast switching times (8.4 s/3 s for the bleaching/coloration processes). Remarkably, the aqueous electrochromic battery exhibited a compelling energy retrieval of 35.7 mW∙h∙m^−2^, where only 47.5 mW∙h∙m^−2^ was consumed during the round-trip coloration–bleaching process. These findings may open up new directions for the development of advanced net-zero-energy-consumption ECD.

In [4,34,58,97], a hybrid electrolyte was developed based on aluminum trifluoromethanesulphonate (Al(TOF)_3_) and H_3_PO_4_ that could effectively alleviate the passivation, and which exhibited superior stability. Additionally, an ex situ study revealed that the PANI cathode undergoes a process of cointercalation/deintercalation of Al(H_2_PO_4_^−^)x(TOF^−^)y +(H_2_O)n, TOF^−^, and H^+^ during the charging/discharging process, with high reversibility and stability. As a proof of concept, an electrochromic Al//PANI battery was fabricated that combined both electrochromism and energy storage and delivered a higher coloration efficiency of 84 cm^2^ C^−1^ at a wavelength of 630 nm.

### 4.5. Counter Electrode

The counter electrode provides ions, which, depending on the polarity of the applied voltage, are injected into or extracted from the electrochromic coating. The counter electrode should be transparent, with high conductivity, in order to reduce the voltage drop and prevent side reactions. Counter electrodes may include EC films, such as WO_3_/PANI films [98], switching from transparent to blue.
(7)$$\underset{\left(transparent\right)}{\underset{︸}{WO_{3}+PANI+xM^{+}A^{-}}}\overset{}{⇌}\underset{\left(colored\right)}{\underset{︸}{MxWO_{3}+\left(PANI\right)A_{x}}}$$
where x is the number of cations (M^+^, H^+^) and anions (A^‒^, SO_4_^‒^).

Thus, thin-film electrodes broaden the ECD color palette and strengthen the electrochromic effect.

## 5. WO_3_ Film Fabrication

The EC WO_3_ layer is obtained as a thin film on a conductive substrate with an FTO or ITO electrode. There are several WO_3_ fabrication techniques [99] (Figure 23), including magnetron sputtering [100], electrochemical deposition [101,102,103,104,105,106], spray pyrolysis [107], sol–gel [108,109], mechanical sputtering [110,111], etc. These technologies are based on electrochemical, chemical and physical principles. C. G. Granqvist [32] provided a comprehensive survey of WO_3_ fabrication technologies.

Table 7 shows a comparative analysis of WO_3_ fabrication technologies.

The majority of technologies shown in Figure 24 are currently in use at the time of writing. Optical contrast is a key parameter for evaluating EC device quality. However, nowadays, there is no universal method that would satisfy all modern requirements. Each method has its own advantages and shortcomings.

WO_3_ film characteristics include porosity, crystallinity and crystal size; these properties are highly dependent on manufacturing conditions and on production technology. The requirements for WO_3_ thin films include uniformity, low production cost, and long life cycle. Unfortunately, the production of a uniform WO_3_ film with good adhesion still remains a problem.

The vacuum deposition method makes it possible to obtain high-density WO_3_ films on a large flat surface, and the thickness and composition can be controlled during the deposition process [100]. Vacuum-deposited WO_3_ films have an amorphous structure, and annealed WO_3_ films have a crystalline structure. However, these technologies are highly expensive due to the expensive equipment. Many glass manufacturing companies still prefer vacuum deposition technologies, regardless of the cost, because WO_3_ films obtained by vacuum deposition are stable, reliable and adjustable.

Chemical vapor deposition (CVD) is used for depositing WO_3_ films on a substrate [112]. However, during the deposition process substrates are heated to a high temperature, which can lead to structural changes in the conductive layer. Electron beam evaporation technology is a well-known method for preparing electrochromic WO_3_ films [113,114].

### 5.1. Electrochemical Deposition

Electrochemical deposition (electrodeposition) is a method of low-temperature synthesis of WO_3_ films. Figure 25 and Figure 26 show a three-electrode system in which conductive FTO or ITO electrodes serve as a working electrode and a platinum electrode is used as a counter electrode.

The applied potential is shown relative to the reference electrode. The most common reference electrode is the silver/silver chloride (Ag/AgCl) electrode (Figure 26), due to the stability of the electrode potential.

The mechanism of electrochemical deposition of electrochromic WO_3_ films has been well investigated [106]; metal or precursor ions are transferred to the working electrode (cathode) under the influence of an applied electrical field. In this case, the metal deposition process can be described by the reaction:(8)$$M^{+}+e^{-}\to M$$

As already mentioned [115,116], the electrochemical deposition method makes it possible to deposit WO_3_ films on large-area conductive substrates. However, special equipment is required for the deposition process. The main advantages of this method include: low cost and fast deposition, while not requiring high-temperature heating and deep vacuum.

### 5.2. Sol–Gel

Colloidal oxide can be synthesized by polycondensation, by acidification of aqueous salt solution, or by hydrolysis of organometallic compounds (Figure 27). Recently, there has been growing interest in the use of the sol–gel process to produce multilayer electrochromic coatings based on non-organic compounds. The main advantage of this reaction is that liquid compounds are converted into solid compounds [117].

Most alkoxides used for electrochromic materials can be produced in several stages [91]:
(1)hydrolysis with the formation of reactive M–OH groups:(9)$$M-\mathrm{OR}+H_{2}O\to M-\mathrm{OH}+\mathrm{ROH}$$
(2)condensation resulting in bridge oxygen formation:(10)M-OH+RO-M→M-M+ROH
(11)$$M-\mathrm{OH}+\mathrm{HO}-M\to M-M+H_{2}O$$


There are different types of sol–gel processes, such as centrifugation, immersion coating and spraying (Figure 28). The sol–gel method, widely applied in material synthesis, is also used to modify the electrode surface [118].

Sol–gel methods make it possible to produce large-area WO_3_ films at lower cost in comparison with traditional vacuum methods [119]. The advantages of this method include: universality of sol–gel processes, easy control of microstructure and composition under low-temperature conditions, relatively simple and inexpensive equipment, control of microstructure, crystal size, porosity and composition of the deposited films, which is important, since these characteristics affect thin film kinetics, durability and staining efficiency [120]. However, many problems still remain to be solved, among them solution stability, large-area uniformity, insufficient adhesion, insufficient film thickness, and low repeatability.

### 5.3. Spray Pyrolysis

The main principle of spray pyrolysis is the pyrolytic decomposition of salt solution sprayed on substrate consisting of deposition target material (Figure 29). The sprayed solution undergoes pyrolytic decomposition and forms a crystallite or a crystallite cluster when the drop comes into contact with the hot substrate surface.

By-products and solvents evaporate during spraying. The hot substrate provides thermal energy for thermal decomposition. After thermal decomposition, sintering and crystallization of the crystallite clusters occur, ultimately leading to film formation. The technique is used for the deposition of dense and porous films on different substrates, such as glass, ceramics and metal.

Spray pyrolysis is a simple and relatively inexpensive method that does not require a vacuum. This method allows large-area uniform films with good adhesion to be produced. Moreover, film properties can be easily modified by changing the spray parameters, such as substrate temperature, flow pressure and the molarity of the precursor solution. The main advantage of this method is that it works at moderate temperatures (100–500 °C) and allows films to be obtained even on low-quality substrates. It offers an easy way of doping films with any elements in any proportion by adding them in some form to the spray solution [121,122]. In [123], V_2_O_5_-WO_3_ composite films were reported to exhibit high coloration efficiency (49 cm^2^ ∙C^−1^). Ref. [124], a fibrous reticulated WO_3_ film obtained by pulsed spray pyrolysis was reported to have a coloration efficiency of 34 cm^2^ ∙C^−1^ at λ = 630 nm.

Spray pyrolysis is a cost-effective method for obtaining highly adhesive homogeneous WO_3_ films with different microstructures. The technology can also be used to produce multilayer films, which is achieved by varying the spray composition. However, this method also has disadvantages, such as the non-uniformity of films, large grain size due to uncontrolled sputtering, solvent loss, and low deposition rate. The mentioned advantages of the spray pyrolysis method make it suitable for industrial applications.

### 5.4. Magnetron Sputtering

Magnetron sputtering is a deposition technology defined as “cathodic sputtering of target material in magnetron discharge plasma (crossed field discharge)”, and is shown in Figure 30.

In this process, permanent magnets are arranged below the target plate so as to produce a magnetic field close to the target material. This concentrates the electrons and causes them to travel in a spiral fashion along the magnetic flux lines of the target instead of wandering around the target material [100].

Magnetron sputtering is the most up-to-date deposition technology [99,100], and is widely used in the industrial and scientific spheres. The frequency of the applied positive DC voltage varies from 20 to 350 kHz, while reversed pulse duration is dependent on dielectric surface discharge [125]. Negative voltage usually varies by an amount equivalent to 10% of the average positive voltage. When the duration and number of positive voltage pulses are sufficient to create electric current, the target surface is bombarded with ions, and when the voltage becomes negative, the incoming ions are repelled. Chen [126,127] investigated WO_3_ films deposited by pulsed magnetron sputtering at a constant frequency of 70 kHz; the O_2_/Ar ratio was reported to vary from 0.2 to 1.0.

The disadvantages of this method include the expensive equipment required and the high energy intensity, which significantly increases ECD cost. The magnetron sputtering technique is used to produce FTO or ITO electrodes on transparent surfaces.

## 6. Nanomaterials for Electrochromic Devices

ECW control the transmittance of light and solar radiation by changing their optical transmittance (transparent, semitransparent and colored states), which ensures comfortable indoor environments and makes it possible to achieve energy savings in buildings. Recent advances in ECD technology emerging in the 1970s led to the creation of different types of ECD. However, there are still problems with respect to the commercialization of EC devices, including aspects such as their high production cost [99], the stability of their long-term operation, and the production of uniform electrochromic films to provide uniformity of coloration in large-area ECW [28,29,30,31,32]. Nanotechnologies can be efficiently used to produce low-cost high-performance ECD [128].

In [129], an experiment was described in which reduced graphene oxide (rGO) films were electrodeposited on indium tin-oxide-coated polyethylene terephthalate substrates (ITO-PET) from graphene oxide (GO), and the resulting flexible transparent electrodes were used in ethyl viologen (EtV2^+^) electrochromic devices. During continuous testing, the resulting devices, which contained GO/rGO in the electrochromic mixture, exhibited a lower switching voltage between the colored and bleached states. Graphene oxide (GO) and reduced graphene oxide (rGO) enabled devices with higher optical contrast than those free of GO at the same applied voltage.

In [130], WO_3_/rGO nanocomposite film was fabricated by sol–gel centrifugation using a mixed colloidal dispersion of WO_3_ precursor and GO. It was reported that the WO_3_/rGO nanocomposite film exhibited shorter coloration and bleaching times (T_c_ = 9.5 s and T_b_ = 7.6 s), higher coloration efficiency (75.3 cm^2^ ∙C^−1^ at 633 nm), larger optical modulatory range (59.6% at 633 nm) and better cyclic stability compared with WO_3_ films; these advantages were attributed to faster Li^+^ ion diffusion and electron transfer rate.

Optically adjustable electrochromic films are basic and important components of electrochromic devices; therefore, the performance of EC devices is strongly dependent on EC film structure, morphology and fabrication method [131].

Amorphous WO_3_ films have a porous structure. Crystalline WO_3_ exhibits better durability compared to amorphous WO_3_, due to its denser structure and low dissolution rate (stability in acidic solution is less than 4 pH) [93,94,132]. However, crystalline WO_3_ possesses high bulk density, which increases switching time and reduces coloring efficiency, so nanostructured WO_3_ with a large specific surface area is expected to have a faster response time and a good durability. Recently, publications have appeared [105,133] on the use of nanoscale or nanoporous WO_3_ (Figure 31) that exhibit fast switching speed and high coloration efficiency due to possessing a good and suitable band gap (~2.6 eV). In [134,135,136] the technologies for producing nanostructured WO_3_ films are discussed (Figure 32).

Nanocrystal-in-glass WO_3_ thin films are considered to be the most promising cathodic electrochromic material [113]. In [137], an all-solution technology was developed for large-area low-cost preparation of electrochromic films. A WO_3_/ITO dispersion was successfully developed; high-electrical-conductivity ITO nanoparticle networks along with ITO coating on glass were able to serve as extended 3-dimensional electrodes, forming a microelectrical field and acting as the pathways for electron diffusion to WO_3_ nanorods. In [138], h-WO_3_ QDs with an average size of 1.2 nm were successfully prepared by a simple decomposition process of tungsten acid in ethylene glycol.

At present, various interactions have been introduced at the interface between the organic and inorganic phases. The expected improved electrochemical and electrochromic performances of the nanocomposites have been obtained. Among of these interactions, covalent bonds have the strongest interaction, although their preparation is relatively complicated [131,138].

Thus, it is an important first step for the fabrication of inexpensive EC “Smart Windows”, and should shape the future research on solution-based processes.

## 7. Conclusions

It was possible, within the scope of this article, to provide a comprehensive review of the large area of new electrochromic materials, and the authors had to use their discretion in choosing up-to-date findings to illustrate this exciting area.

In summarizing this review of the literature on electrochromism in electrochromic materials, and in WO_3_ films in particular, the following conclusions can be drawn:(1)There are several hypotheses concerning the mechanism of electrochromism in WO_3_. Generally, the electrochromic effect in WO_3_ films can be described as an electrochemical cathodic polarization during which H^+^ ions are transferred from the electrolyte and an electron is transferred from the ITO electrode. As a result, WO_3_ film switches from a bleached to a colored state; its color varies from pale blue to dark blue and black. The conductivity of WO_3_ films is determined by the presence of cations (H^+^, Li^+^, etc.) and electrons. As already mentioned, the coloration mechanism in WO_3_ films has still been insufficiently investigated.(2)Despite a large number of works devoted to the study of electrochromic WO_3_ films, the influence of the structural state on optical properties during the electrochemical reaction has not been fully investigated. Different film deposition techniques have been proposed. Film morphology is dependent on deposition technique and can be amorphous, crystalline, nanocrystalline or hybrid. Additionally, there is still a constant need for new technologies to produce WO_3_ films, and nanostructured WO_3_ films in particular. Therefore, there is a necessity to study the fabrication of amorphous, crystalline and nanocrystalline WO_3_ films, including their GO/rGO modification. Analysis of literary sources makes it possible to identify prospects for the development of WO_3_/rGO fabrication technologies. The obtained data will be useful in the development of WO_3_ fabrication technologies.

Today, the development of the energy-efficient glazing sector is impossible without EC Modern nanomaterials make ECD an interesting commercial product that has obvious advantages over its competitors, such as PDLC, LCD and SPD. In this regard, according to some forecasts, the market for electrochromic “Smart Window” will expand in the next 5–7 years. First of all, thanks to the development of modern technologies and nanomaterials, as well as intensive research into EC by companies and scientific laboratories around the world.

## Figures and Tables

**Figure 1 nanomaterials-11-02376-f001:**
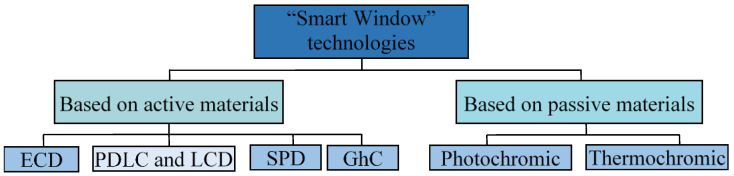
“Smart Windows” classification.

**Figure 2 nanomaterials-11-02376-f002:**
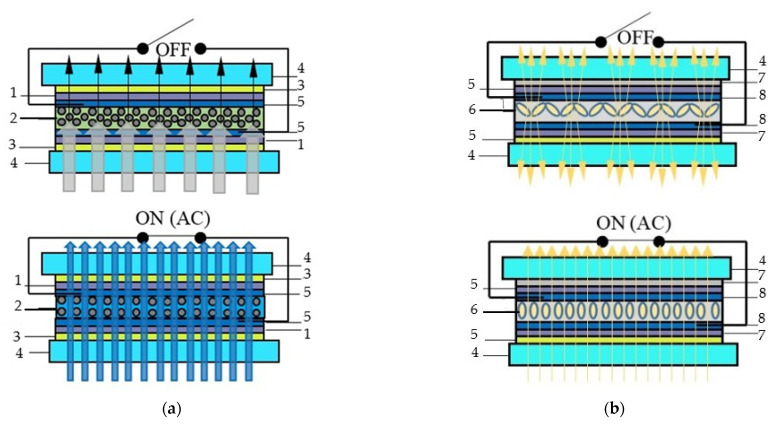
“Smart Windows” SPD and PDLC technology sandwich structures and operating principles: (**a**) SPD: off—light modulation mode, on—transparent mode; (**b**) PDLC: off—semi-transparent mode, on—transparent mode; 1—retaining film; 2—suspended particle; 3—adhesive layer; 4—glass; 5—conductor; 6—liquid crystal layer; 7—interlayer film; 8—liquid crystal.

**Figure 3 nanomaterials-11-02376-f003:**
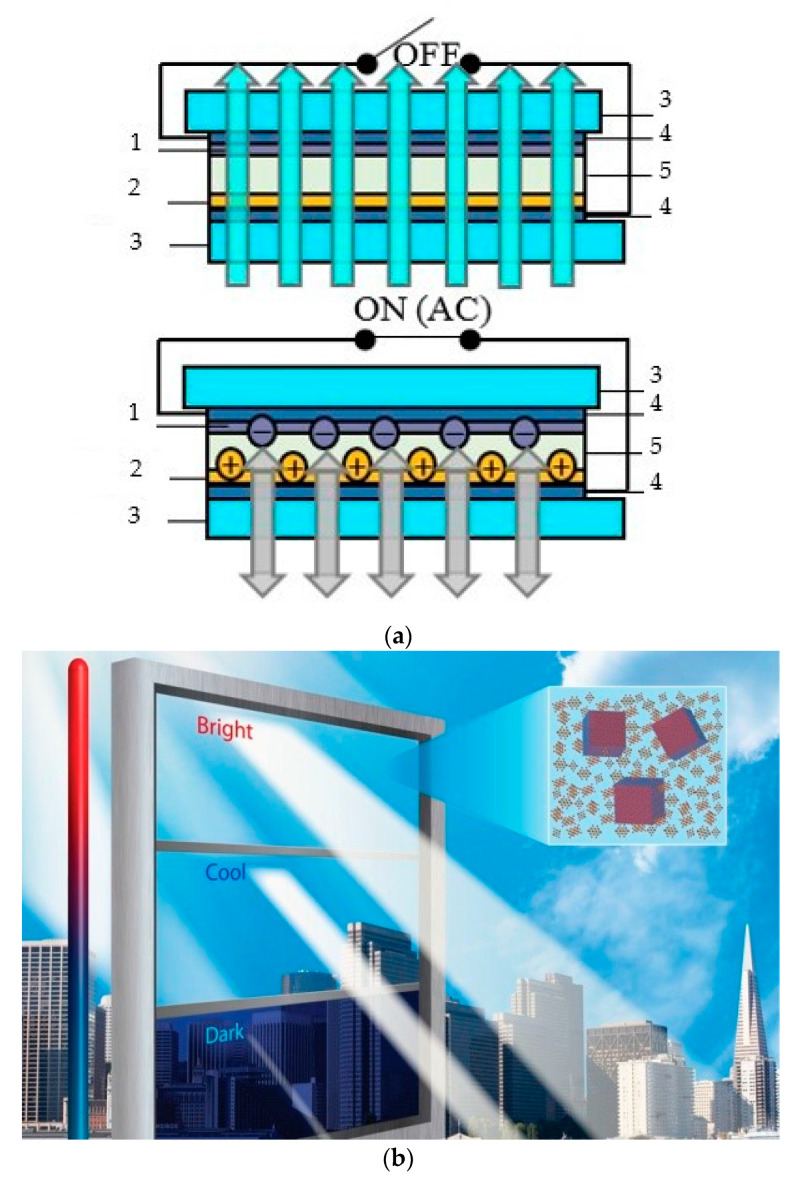
“Smart Window” electrochromic technology: (**a**) sandwich structure and operating principle (bleached state): transmitted and reflected light modulation: 1—electrochromic layer; 2—ion storage layer; 3—glass; 4—conductive layer; 5—ion conductor/electrolyte; (**b**) total view.

**Figure 4 nanomaterials-11-02376-f004:**
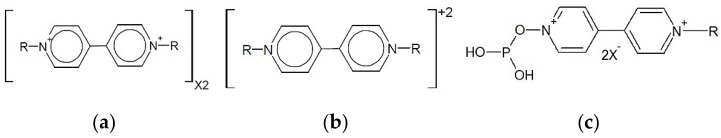
Viologen: (**a**) general chemical formulae of viologen; (**b**) viologen ion; (**c**) general structure for viologens modifying the titania surface.

**Figure 5 nanomaterials-11-02376-f005:**
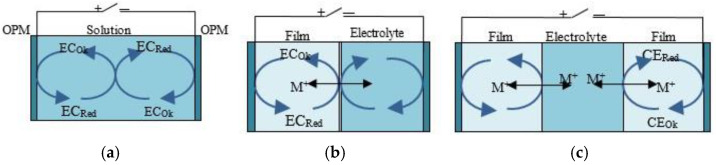
Types of ECW: (**a**) type I (solution); (**b**) type II (hybrid); (**c**) type III (battery-powered); EC—electrochromic layer; CE—counter-electrode layer; TCO—inorganic oxide.

**Figure 6 nanomaterials-11-02376-f006:**
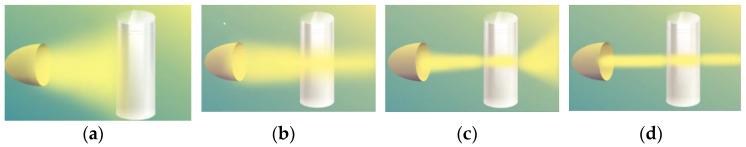
Interaction of radiation with an EC: (**a**) reflection; (**b**) absorption; (**c**) dispersion; (**d**) transmittance.

**Figure 7 nanomaterials-11-02376-f007:**
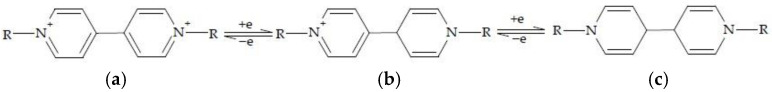
Three general viologen redox states (in terms of electron transfer): (**a**) dication; (**b**) radical cation; (**c**) neutral state.

**Figure 8 nanomaterials-11-02376-f008:**
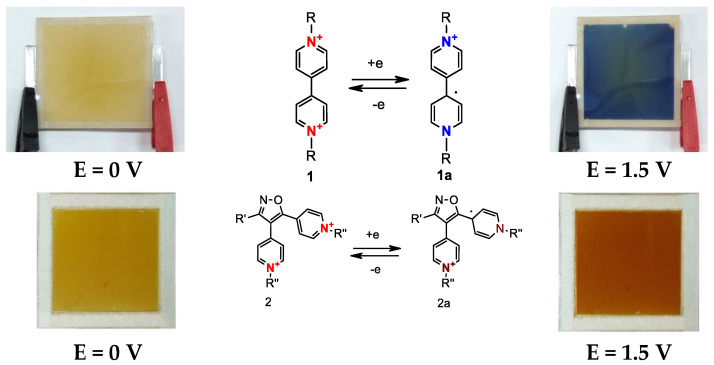
Electrochromic transition cycle.

**Figure 9 nanomaterials-11-02376-f009:**
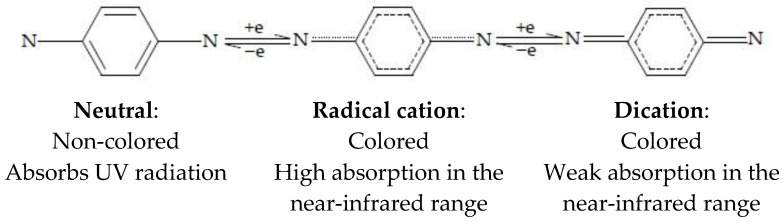
Redox chemistry of Phenylenediamine (Wurster’s blue), description of optical behavior in redox states.

**Figure 10 nanomaterials-11-02376-f010:**
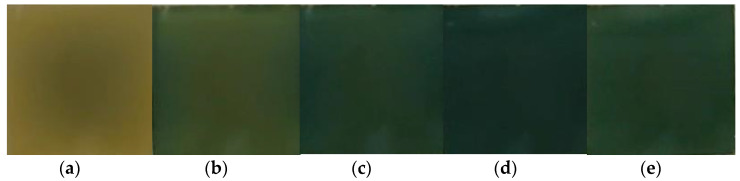
New electrochromic compounds obtained by reactions involving the cycloaddition of nitrile oxides to 1,2-bis (4-pyridinyl) ethylene derivatives (electrochromic transition): (**a**)—neutral state; (**b**–**e**)—oxidized states.

**Figure 11 nanomaterials-11-02376-f011:**
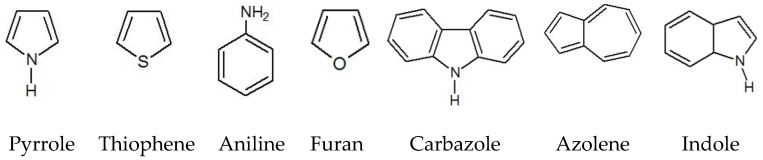
Molecules of heterocyclic aromatic compounds.

**Figure 12 nanomaterials-11-02376-f012:**
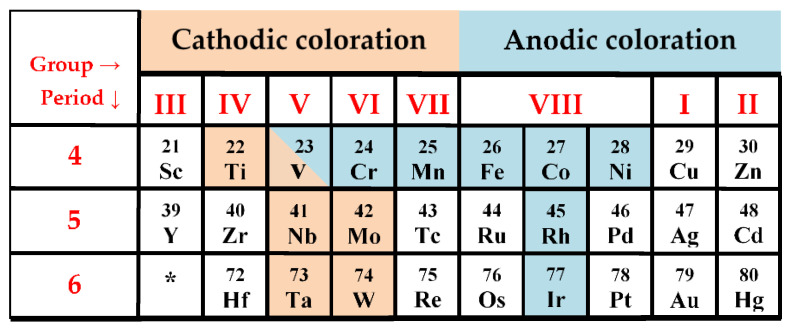
Electrochromic transition metal oxides. *—lantanoids.

**Figure 13 nanomaterials-11-02376-f013:**
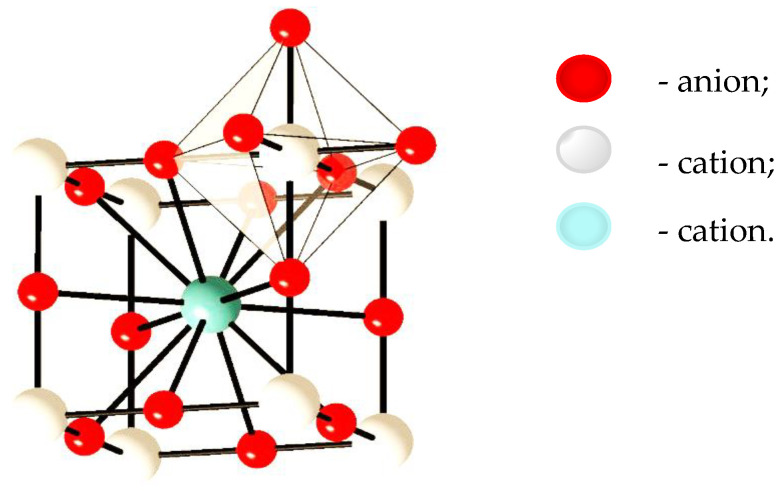
Crystal structure of MeO_6_ perovskite.

**Figure 14 nanomaterials-11-02376-f014:**
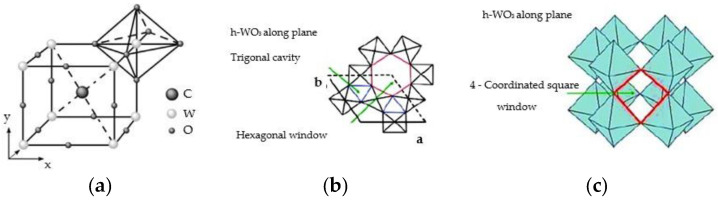
Crystal structure of CWO_3_ perovskite: (**a**) general view; (**b**) h-WO_3_ along plane; (**c**) h-WO_3_ along plane.

**Figure 15 nanomaterials-11-02376-f015:**
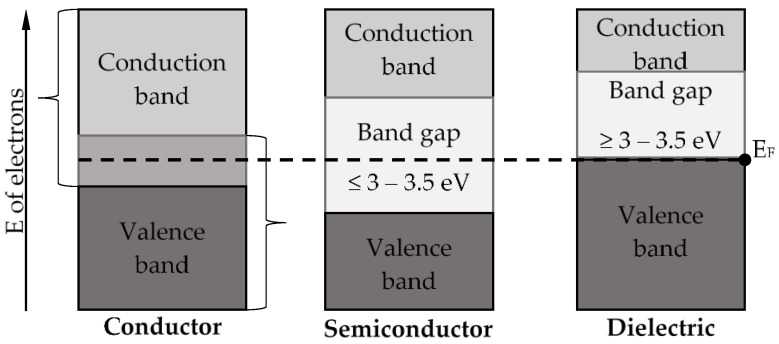
Classification of materials by conductivity (according to zone theory).

**Figure 16 nanomaterials-11-02376-f016:**
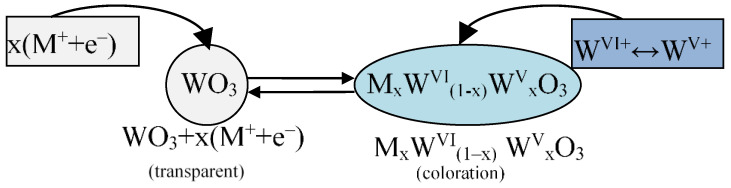
Electrochromic mechanism of WO_3_ film.

**Figure 17 nanomaterials-11-02376-f017:**
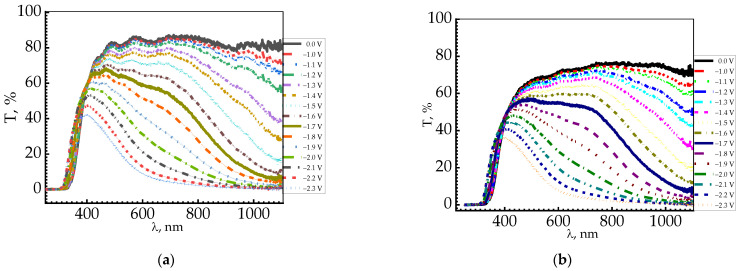
Optical transmission spectra of WO_3_ film during electrochromic response obtain by electrochemistry (cathodic) deposition: (**a**) WO_3_ at constant potential; (**b**) WO_3_/GO deposition at AC potential [77].

**Figure 18 nanomaterials-11-02376-f018:**
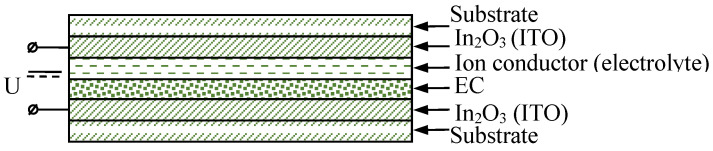
ECD structure.

**Figure 19 nanomaterials-11-02376-f019:**
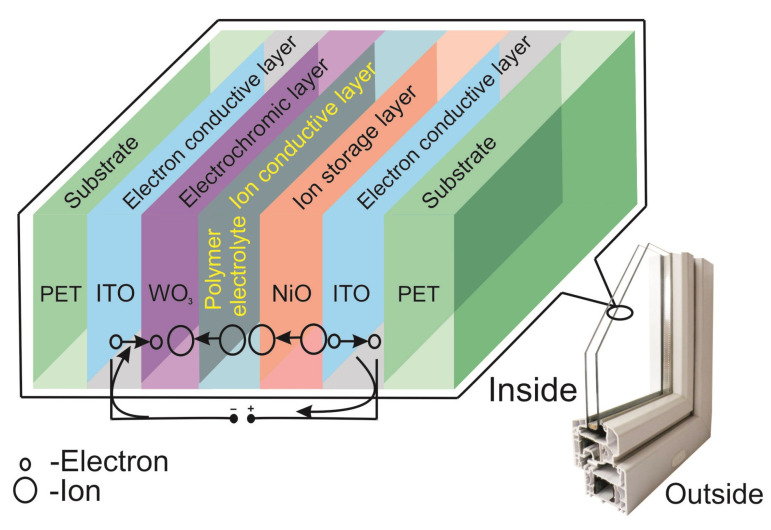
ECW scheme showing voltage-induced transfer of positive ions and electrons to transparent conductive layers.

**Figure 20 nanomaterials-11-02376-f020:**
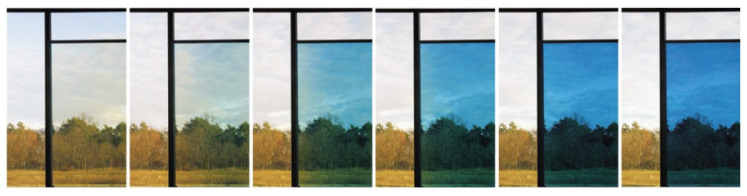
ECW color cycle (colored ↔ semitransparent ↔ transparent state).

**Figure 21 nanomaterials-11-02376-f021:**
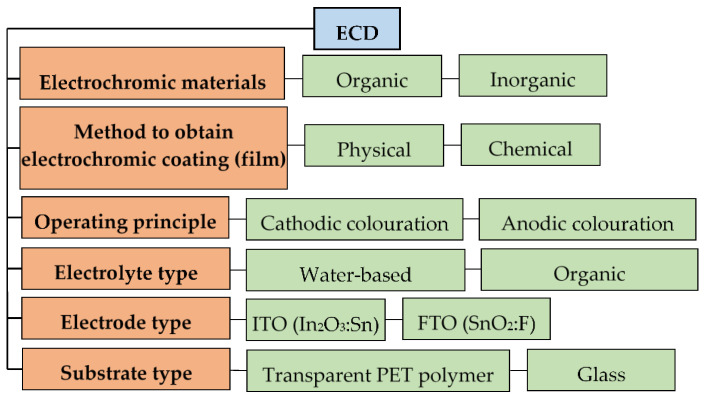
ECD classification.

**Figure 22 nanomaterials-11-02376-f022:**
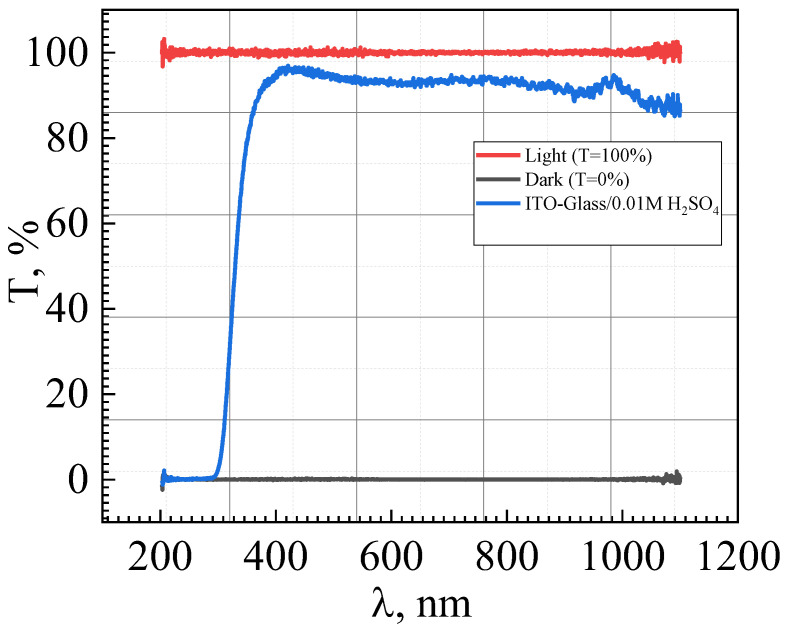
Visible and near-infrared transmission spectra of WO_3_-ITO-glass.

**Figure 23 nanomaterials-11-02376-f023:**
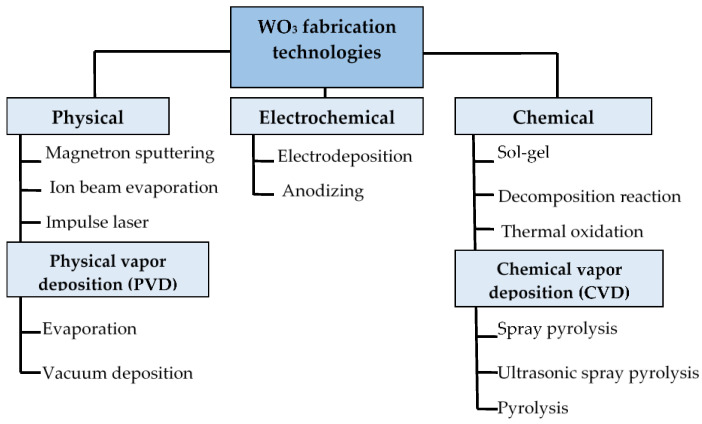
Classification of WO_3_ fabrication technologies.

**Figure 24 nanomaterials-11-02376-f024:**
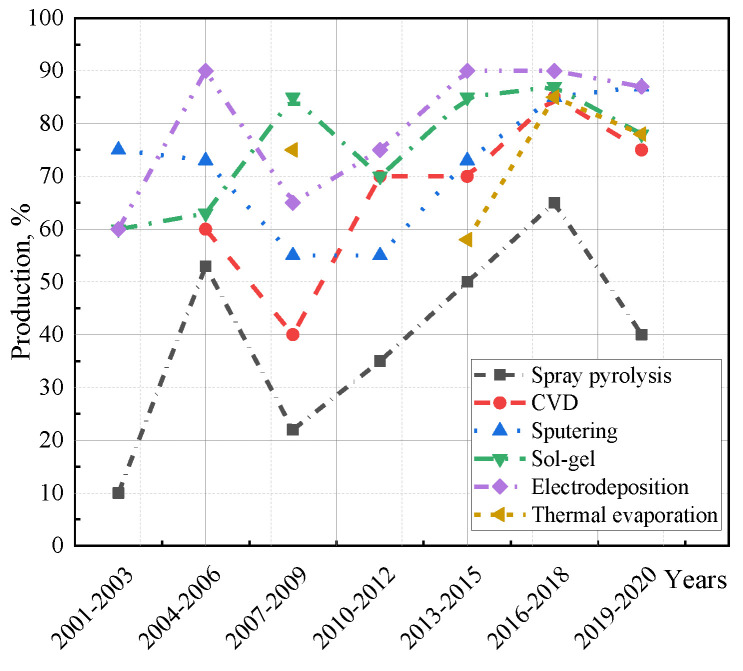
Contrast response curves for WO_3_ films obtained by different processes during the reporting period.

**Figure 25 nanomaterials-11-02376-f025:**
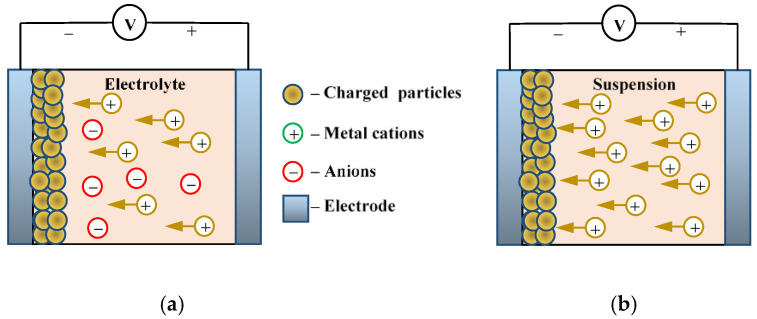
Two types of electrodeposition processes: (**a**) electroplating; (**b**) electrophoretic deposition.

**Figure 26 nanomaterials-11-02376-f026:**
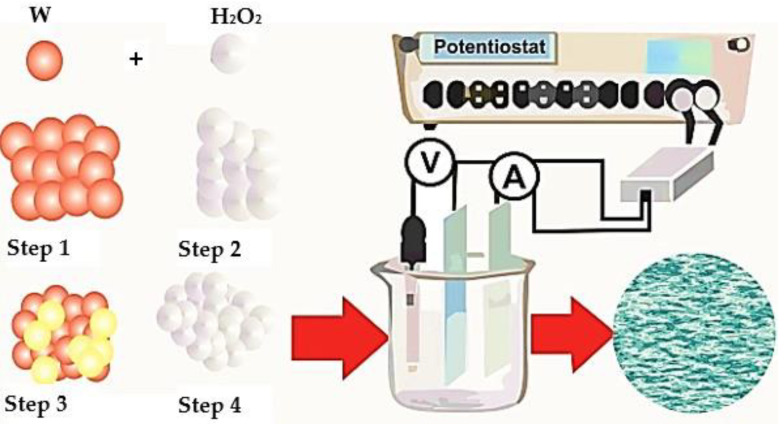
Growth mechanism of electrodeposited WO_3_ film.

**Figure 27 nanomaterials-11-02376-f027:**
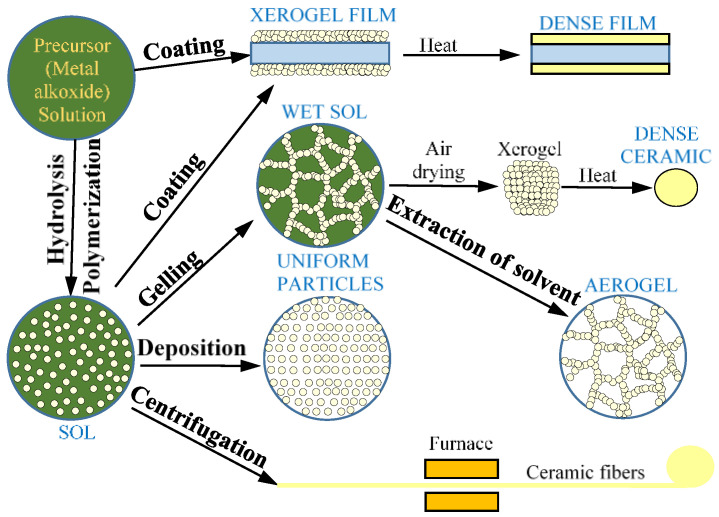
Sol–gel process scheme.

**Figure 28 nanomaterials-11-02376-f028:**
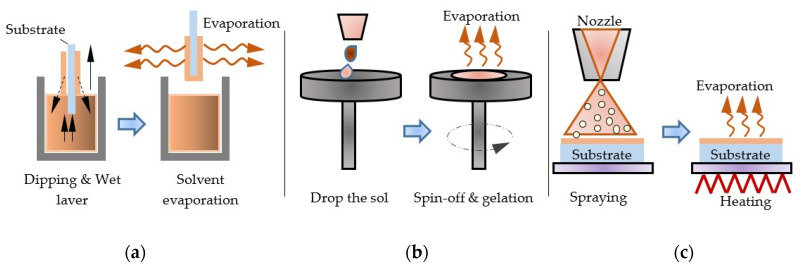
Types of sol–gel processes: (**a**) immersion coating; (**b**) centrifugation; (**c**) spraying.

**Figure 29 nanomaterials-11-02376-f029:**
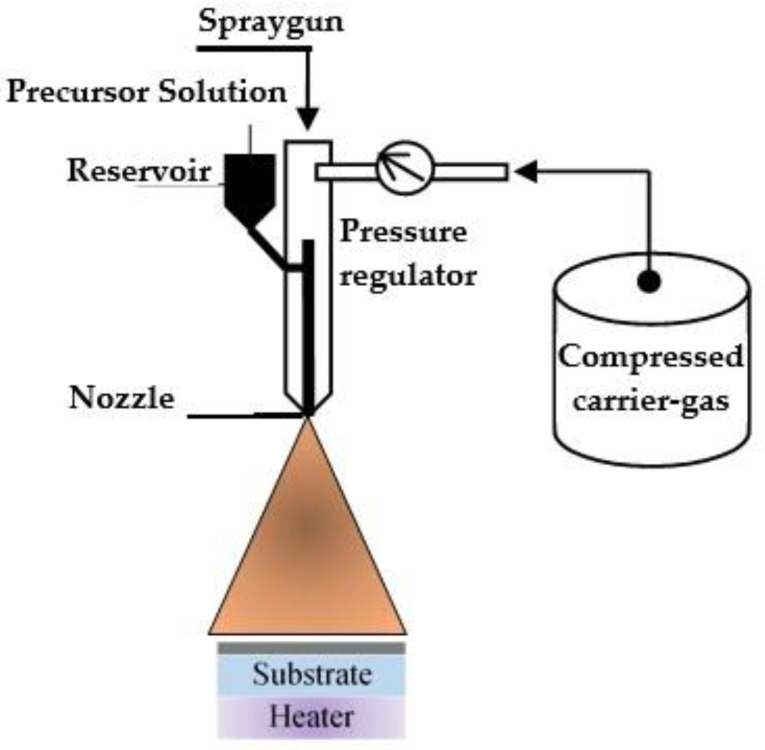
Pyrolytic deposition of EC films.

**Figure 30 nanomaterials-11-02376-f030:**
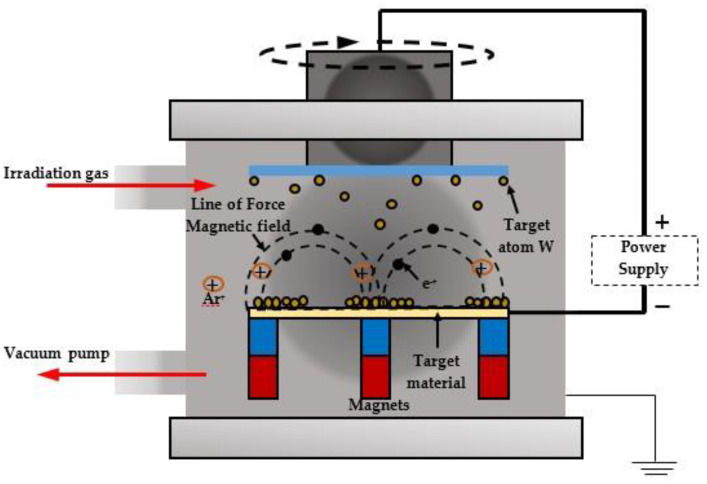
Magnetron sputtering apparatus (working principal).

**Figure 31 nanomaterials-11-02376-f031:**
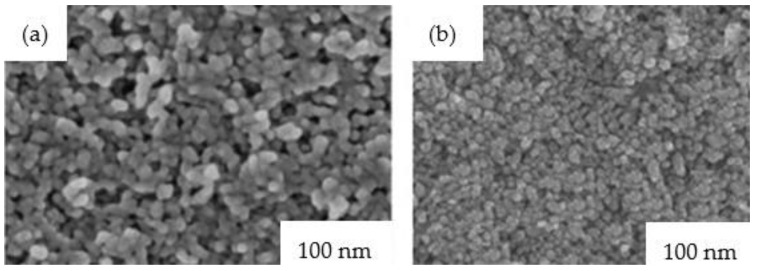
FE-SEM micrographs for nc-TiO_2_ nanoparticles film: (**a**) before deposition; (**b**) deposited on H_2_W_2_O_11_ electrolyte surface.

**Figure 32 nanomaterials-11-02376-f032:**
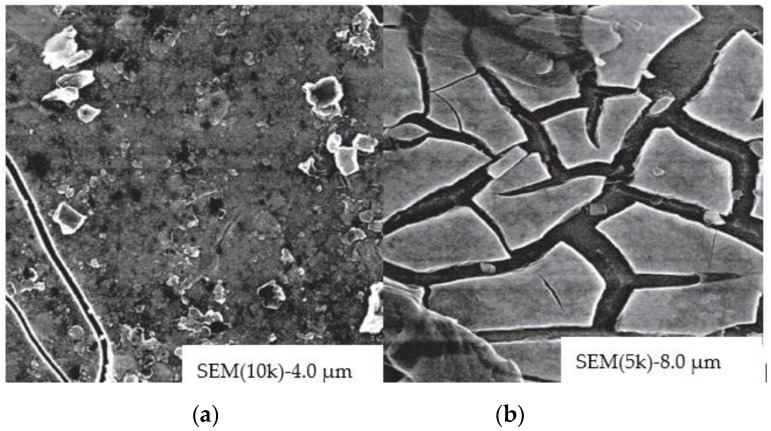
Nanostructured films obtained by electrochemical deposition: (**a**) WO_3_(GO_2 mL_); (**b**) WO_3_(GO_1 mL_) [135].

**Table 1 nanomaterials-11-02376-t001:** Comparison of “Smart Window” technologies.

Technology	Energy Efficiency, W/m^2^	Energy Saving, W/m^2^ (Energy Saving in Building)	Transparency, %	Modulation Time, s	Cost, (c.u./m^2^)
ECW	+	+	+	–	–
SPD	–	–	+	+	+
PDLC	–	–	+	+	+
LCD	–	–	+	+	+

**Table 2 nanomaterials-11-02376-t002:** General classification of EC.

EC Class	Chemical Name	Application	Ref.
**Organic**			
Conductive polymers	PEDOT (where EDOT = C_6_H_6_O_2_S), PPy (where Py = Pyrrole = C_4_H_5_N), PT (where T = thiophene = C_4_H_4_S), PANI (where ANI = aniline = C_6_H_4_S)	“Smart Windows”, displays	[13,33]
Viologens	3-aryl-4,5-bis (pyridine-4-yl) isoxazole derivatives	Antiglare mirrors and displays	[21,28]
Transition metals and lanthanoids	poly [Ru^II^(vbpy)_2_(py)_2_]Cl_2_ (being py = pyridine = C_5_H_5_N)	Smart mirrors	[26,30]
Metal phthalocyanines (Pc)	[Lu(Pc)_2_] being Pc = C_32_H_18_N_8_ et al.	Displays	[7,30]
**Inorganic**			
Transition metal oxides (TMOs)	WO_3_, MoO_3_, V_2_O_5_, TiO_2_ Nb_2_O_5_, Ir(OH)_3_, NiO et al.	“Smart Windows”, antiglare mirrors	[32,34]
Prussian blue (PB)	Prussian blue (C_18_Fe_7_N_18_), Prussian brown (C_6_Fe_2_N_6_), Prussian green (C_3_FeN_3_), Prussian white (C_6_Fe_3_N_6_)	“Smart Windows”, displays	[7,29]

**Table 3 nanomaterials-11-02376-t003:** Classification of EC materials (according to I. F. Chang [36]).

EC Type	EC Material	Electrochromic Reaction Mechanism	Application	Ref.
I (solution)	(1) Methylviologene (MV, 1, 10-dimethyl-4,4′-bipyrindnium, 3-aryl-4,5-bis (pyridine-4-yl) isoxazole; (2) Phenothiazine (C_12_H_9_NS) in non-aqueous solution	MV^2+^ + e^−^_(bleached)_↔MV^+●^_(colored)_	Night vision systems, mirrors	[37,38]
II (hybrid)	(1) Cyanophenylparaquate (CPQ, 1-1 cyanophenyl-4,4′-bipyridine, paraquat = C_12_H_14_Cl_2_N_2_, otherwise known as viologen, due to the herbicide name) in aqueous solution (2) Heptyl or benzylviologene (HV or BzV) or methoxyfluorene compounds C_3_H_4_Cl_2_F_2_O in acetonitrile solution (C_2_H_3_N)	CPQ^2+^ + e^−^ + X^−^↔[CPQ^+●^X^−^]	Electrochromic paper, “Smart Window”	[39,40,41]
III (battery-powered)	(1) Almost all inorganic EC materials, such as transition metal oxides: WO_3_, MoO_3_, V_2_O_5_, TiO_2_ Nb_2_O_5_, Ir(OH)_3_, NiO; (2) Phthalocyanine (Pc = C_32_H_18_N_8_) (3) Metal complexes and hexacyanometallates, such as Prussian blue (PB = C_18_Fe_7_N_18_) (4) Conductive polymers: polypyrrole (PPy), polythiophene (PT), polyaniline (PANI)	MO_y_ + x(H^+^ + e^−^)↔H_x_MO_y(colored)_	“Smart Window” (Boeing 757), Electro-chromic paper	[32,42,43]

**Table 4 nanomaterials-11-02376-t004:** Conductive polymers obtained by the oxidation of monomeric aromatic compounds (neutral and oxidized states).

	Organic EC
State	PANI	P3MPy	MEPA	P3MT	PPY	PT
Neutral	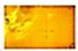	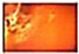	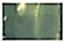	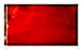	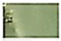	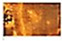
Oxidized	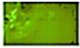	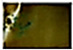	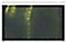	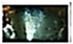	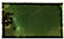	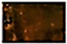

**Table 5 nanomaterials-11-02376-t005:** Color variation in WO_3_, NiO and WO_3_/NiO electrochromic films (colored and bleached states).

	Inorganic EC
State	WO_3_	NiO	NiO/WO_3_
Oxidized	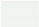	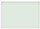		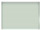	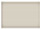		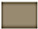		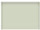	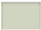		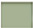
Reduction			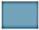		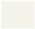	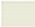	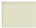				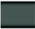	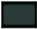

**Table 6 nanomaterials-11-02376-t006:** Electrochemical reactions of certain oxides.

Metal Oxide	Electrochemical Reaction	Color Change	Reaction Type
Manganese oxide (II)	$MnO_{2}+ze^{-}+zH^{+}\Leftrightarrow MnO_{\left(2-z\right)}\left(OH\right)$	Yellow ↔ brown	A
Cobalt oxide (II)	$3CoO+2OH^{-}\Leftrightarrow Co_{3}O_{4}+H_{2}O+2e^{-}$	Green ↔ light blue	A
Nickel oxide (II)	$NiO_{x}H_{y}\Leftrightarrow \left[Ni^{II}\left(1-z\right)Ni^{III}{}_{z}\right]O_{x}H_{\left(y-z\right)}+zH^{+}+ze^{-}$	Colorless ↔ brown	A
Molybdenum oxide (VI)	$MoO_{3}+x\left(Li^{+}+e^{-}\right)\Leftrightarrow Li_{x}Mo^{VI}{}_{\left(1-x\right)}Mo^{V}{}_{x}O_{3}\frac{1}{2}$	Colorless ↔ blue	C
Vanadium oxide (V)	$Li_{x}V_{2}O_{5}\Leftrightarrow V_{2}O_{5}+x\left(Li^{+}+e^{-}\right)$(A) $V_{2}O_{5}+x\left(M^{+}+e^{-}\right)\Leftrightarrow M_{x}V_{2}O_{5}$(C)	Blue ↔ brown (A) Yellow ↔ light blue (C)	C/A
Cerium oxide (IV)	$CeO_{2}+x\left(Li^{+}+e^{-}\right)\Leftrightarrow Li_{x}CeO_{2}$	Yellow ↔ transparent	C
Niobium oxide (V)	$Nb_{2}O_{5}+x\left(Li^{+}+e^{-}\right)\Leftrightarrow Li_{x}Nb_{2}O_{5}$	Colorless ↔ light blue	C
Ruthenium oxide (IV)	$RuO_{2}\cdot 2H_{2}O+H_{2}O+e^{-}\Leftrightarrow 0,5\left(Ru_{2}O_{3}\cdot 5H_{2}O\right)+OH^{-}$	Blue ↔ brown/yellow	C
Indium oxide (ITO)	$In_{2}O_{3}+2x\left(Li^{+}+e^{-}\right)\Leftrightarrow Li_{2x}In^{III}{}_{\left(1-x\right)}In^{I}{}_{x}O_{3}$	Colorless ↔ light blue	C
Iridium oxide (III)	$Ir{\left(OH\right)}_{3}\Leftrightarrow IrO_{2}\cdot H_{2}O+H^{+}+e^{-}$	Colorless ↔ blue/grey	C
Tungsten oxide (VI)	$W^{V}O_{3}+x\left(Li^{+}+e^{-}\right)\Leftrightarrow Li_{x}W_{\left(1-x\right)}^{VI}W_{x}^{V}O_{3}$ $W^{V}O_{3}+x\left(H^{+}+e^{-}\right)\Leftrightarrow H_{x}W_{\left(1-x\right)}^{VI}W_{x}^{V}O_{3}$	Colorless ↔ blue/black	C

**Table 7 nanomaterials-11-02376-t007:** Comparison of three basic approaches to films WO_3_ fabrication.

Technology Types	Scalability	Equipment Cost	Process Costs	Coating Uniformity
Electrochemical	+/−	+	+	+/−
Chemical	+/−	+	+	−
Physical	+	−	−	+

## Data Availability

The data presented in this study are available on request from the corresponding author.

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
