# Peer review of "A Brief Overview of Electrochromic Materials and Related Devices: A Nanostructured Materials Perspective"

_nanomaterials, 2021, doi:10.3390/nano11092376_

Round 1

Reviewer 1 Report

This manuscript summarizes electrochromic materials in terms of structure, preparation, and performance. This review may be worthy of publication as nanomaterials. However, several critical issues with the paper would need to be addressed prior to publication in nanomaterials.

  1. Electron beam evaporation technology is a well-known method for preparing electrochromic WO3 films. I suggest that the authors should add several reports on e-beam evaporated WO3 thin films. [Ref: Inorg. Chem. 2019, 58, 3, 2089–2098; Current Applied Physics 19 (2019) 198–203]
  2. Nanocrystal-in-glass WO3 thin films is considered as the most promising cathodic electrochromic material. The authors should summarize the results and advantages of Nanocrystal-in-glass WO3 thin films. [Ref: nature 2013, 500, 323–326; Inorg. Chem. 2019, 58, 3, 2089–2098]
  3. Cycle stability is an extremely important performance of electrochromic devices. In the recent researches [Ref: Journal of Alloys and Compounds 861 (2021) 158534], ECD can obtain a superior long-term cycling stability of over 10,000 cycles. This manuscript is suggested to review some reports on the devices with high long-term stability.
  4. In page 11, the authors classify electrochromic materials according to band gap. What effects do the conductivity of the materials have on electrochromic performance?
  5. The electrochemical reaction at the WO3/electrolyte interface plays an imperative role in the electrochromic performance of WO3 electrodes. This review is suggested to track some recent reports on the technologies used to investigate interface of materials. [Ref: Scripta Materialia 203 (2021) 114090]

6.In page 11 Table 6, there is a mistaken. “OH+” should be corrected to “OH-”.

  1. In terms of ECD structure, “complementary” and “all-in-one” are the two most common ECD structures. In my opinion, related reports [for example Refs: Journal of Alloys and Compounds 861 (2021) 158534; ACS Appl. Mater. Interfaces 2020, 12, 24, 27526–27536] can be added to help readers better understand the ECD structure.
  2. This manuscript is also suggest track several reports on multivalent conductive ionic (Zn2+, Al3+) electrolyte-based ECD. [Refs: Nanoscale Horiz., 2020, 5, 691-695; Energy Storage Materials 39 (2021) 412–418]
  3. The manuscript has some spelling errors. For example (page 1 line 44) “The functional materials has…”; (page 3 line 94) “Thus, ECW create…” etc.

Reviewer 2 Report

The manuscript presented by Shchegolkov, Jang, Shchegolkov, Rodionov, and Sukhova, entitled "Modern electrochromic materials: structure, properties and methods obtain (a general review)" is a review on a very interesting topic.

Although, I think that the manuscript has to be strongly improved. I would strongly encourage the authors to spend some effort in re-phrasing sentences and prepare more clear figure and table captions.

I have some points:

The first 3 points are examples of the English of the manuscript that needs extensive editing:

1) Title "Modern electrochromic materials: structure, properties and 2 
methods obtain". I do not understand the meaning of "methods obtain". Do the authors mean "methods to obtain such materials"?

2) First sentence of the introduction: "Negative effects of modern technology, such as atmospheric pollution, global warming and reduction fossil resources". The sentence misses the principal verb. In this form, it is not meaningful.

3) Line 47: "Electrochromic materials (EC)  ‒ materials that are able to change colour under the influence of an electric field." Again, te principal verb of the sentence is missing.

Other points:

4) Words within Figure 2 (close to different geometries) are too small.

5) Table 1: the authors use (I assume) "+" and "-" to displayes pros and cons of a certain technology. This should be specified. Moreover, I would suggest to be a bit more quantitative.

6) All the figures and tables need more exhaustive and clear captions that better explain the content of either the figure(s) or the table.

7) Optional: the tables that report data of materials could include the most relevant literature that report such data

8) Figure 30: the magnetron apparatus needs to be explained in better detail, especially in the figure caption. There are a lot of acronyms in the figure that are not explained.

9) In general, I suggest to authors to double check all the acronyms that they have introduced in the manuscript and to explain the meaning of such acronyms. Since the number of acronyms introduced along the manuscript is big, I would suggest to prepare a list of acronyms that can be included at the beginning of the manuscript.

The bibliography is proper and includes the most significant contributions in the field.

Reviewer 3 Report

The manuscript deals with an interesting field of investigation but fails in the definition of a precise focus. The field of EC materials is really large and cannot be properly reviewed in a single paper. Several materials (organic and inorganic) are not even cited in this work and a lot of recent studies and reviews are not reported in the reference section. 

Probably, the authors should narrow the scope of the manuscript, reaching - certainly - a greater degree of completeness.

There are many typos and errors: an extensive correction of English language is required. 

Round 2

Reviewer 1 Report

The topic of the review manuscript on electrochromism and devices is certainly of interest.  The article describes modern electrochromic materials, proposes and classification,  analyzes current tendencies in the development of electrochromic devices, as well as methods for their preparation.
However, there are some questions that need to be resolved before published, as noted below.
1. Notes on the molecular formula of organic materials,such as, pyridine = C5H6M,,,in Table 3.
2. If the elctrochromic anode materials in the article are also summarized, the review paper is more perfect.

Reviewer 2 Report

Shchegolkov, Jang, Shchegolkov, Rodionov Sukhova, and Semenovich have submitted a revised version of the manuscript entitled "Review in Electrochromic Materials and related Devices: organic and inorganic". Together with the revised manuscript, they have sent a response letter in which all the changes and answers to my comments have been reported.

I have read the response letter and the revides manuscript and I believe that the changes and revisions are proper and the manuscript has been significantly improved.

In general, the work is a significant contribution, well organized and comprehensive. The work is scientifically sound and the references are appropriate.

I suggest to accept themanuscript for publication.

Author Response

Thank you very much for your appreciation

Reviewer 3 Report

Authors claim to review materials and devices (organic and inorganic) but the topic is too large for a single review if one considers books from Mortimer or Granqvist, concerning decades of research in such fields. According to the contents of their paper, the authors should focus on a more specific topic, but in an exhaustive way. 

The title should be rearranged properly.

Please check grammar in line 59 and throughout the manuscript: several typos are still present.

Please improve quality of Figure 2 and 3. Are those pictures cited from other papers or prepared by the authors?

Figures from other papers should report their references in the captions.

Tables should be formatted, according to the journal's template.
